# Evolution of histone 2A for chromatin compaction in eukaryotes

**Benjamin R Macadangdang[1,2†], Amit Oberai[1†], Tanya Spektor[1†], Oscar A Campos[1,2], Fang Sheng[1], Michael F Carey[1,2], Maria Vogelauer[1], Siavash K Kurdistani[1,2,3,4]***

[1]Department of Biological Chemistry, University of California, Los Angeles, Los Angeles, United States; [2]Molecular Biology Institute, University of California, Los Angeles, Los Angeles, United States; [3]Department of Pathology and Laboratory Medicine, University of California, Los Angeles, Los Angeles, United States; [4]Eli and Edythe Broad Center of Regenerative Medicine and Stem Cell Research, David Geffen School of Medicine, University of California, Los Angeles, Los Angeles, United States

**Abstract** During eukaryotic evolution, genome size has increased disproportionately to nuclear volume, necessitating greater degrees of chromatin compaction in higher eukaryotes, which have evolved several mechanisms for genome compaction. However, it is unknown whether histones themselves have evolved to regulate chromatin compaction. Analysis of histone sequences from 160 eukaryotes revealed that the H2A N-terminus has systematically acquired arginines as genomes expanded. Insertion of arginines into their evolutionarily conserved position in H2A of a small-genome organism increased linear compaction by as much as 40%, while their absence markedly diminished compaction in cells with large genomes. This effect was recapitulated in vitro with nucleosomal arrays using unmodified histones, indicating that the H2A N-terminus directly modulates the chromatin fiber likely through intra- and inter-nucleosomal arginine–DNA contacts to enable tighter nucleosomal packing. Our findings reveal a novel evolutionary mechanism for regulation of chromatin compaction and may explain the frequent mutations of the H2A N-terminus in cancer.

***For correspondence:**
skurdistani@mednet.ucla.edu

†These authors contributed equally to this work

**Competing interests:** The authors declare that no competing interests exist.

**Reviewing editor**: James T Kadonaga, University of California, San Diego, United States

## Introduction

Genome size, defined as the haploid DNA content of a cell, has increased as eukaryotes evolved from single-cell species to more complex, multicellular organisms. Within the same evolutionary timeframe, nuclear volume has also increased but at a slower rate than genome size expansion (**Maul and Deaven, 1977**; **Olmo, 1982**). While the ratio of nuclear to cell size has remained essentially constant in eukaryotes (**Cavalier-Smith, 2005**), the disproportional increase in genome size relative to the nuclear volume has required organisms with larger genomes to compact their chromatin to greater extents than organisms with small sized genomes. Indeed there is a positive correlation between genome size and native chromatin compaction as measured by dye incorporation into chromatin (**Vinogradov, 2005**). In most eukaryotes, the genome is organized into chromatin by the repeating nucleosomal structure (**Luger et al., 1997**). The nucleosomes stack and fold into higher order structures, serving to systematically compact the genome (**Lieberman-Aiden et al., 2009**; **Duan et al., 2010**) and to regulate molecular processes that are based on DNA (**Celeste et al., 2002**; **Vogelauer et al., 2002**; **Fischle et al., 2005**; **Kouzarides, 2007**; **Fussner et al., 2011**).

The surface of the histone octamer has 14 DNA interaction sites. Each interaction is mediated by an arginine residue that intercalates into the minor groove of the DNA to stabilize the nucleosomal structure (**Luger et al., 1997**; **West et al., 2010**). Arginine is the most commonly used amino acid for interaction with DNA due to its positive charge and the lower energetic cost compared to lysine for

**eLife digest** There are up to three meters of DNA in a human cell. To fit this length into the cell's nucleus in an organized manner, DNA is wrapped around proteins called histones and then tightly packaged to form a structure called chromatin. The interaction between the histones and the DNA is helped by certain amino acids on the surface of the histones fitting snugly into the DNA molecule.

Plants and animals have genomes that are significantly larger than those of single-celled organisms. However, although genome size has increased gradually during the evolution of complex organisms, the size of the nucleus has not undergone a similar expansion. Large genomes are therefore packaged more tightly than small genomes. However, we do not fully understand how different species evolved the ability to do this.

Now Macadangdang, Oberai, Spektor et al. have compared the histones of 160 species ranging from single-celled microorganisms to plants and animals. This revealed that the amino acids in a particular type of histone—histone 2A—vary according to genome size. Organisms with small genomes use histone 2A proteins with fewer arginine amino acids on their surface than organisms with large genomes.

Further experiments showed that yeast cells engineered to contain arginine-rich histones wind their DNA more tightly; and, in some cases when the chromatin was more compacted, the nuclei were also smaller. On the other hand, removing arginines from histones in human cells cause the chromatin to be loosely packed and the nuclei to be larger than normal. Moreover, chromatin is often abnormally packed in cancer cells and Macadangdang et al. found that many of these cells contained histones with fewer arginines than normal.

Plants, animals, and other eukaryotes have evolved a variety of mechanisms to control how much they compact their chromatin in addition to the way discovered by Macadangdang et al. Future work is now needed to determine how these different mechanisms work together in different species such that the chromatin is compacted to the optimal level.

displacing water when intercalating into the minor groove (*Rohs et al., 2009*). Nucleosomes mediate chromatin compaction through a variety of mechanisms. For instance, nucleosomes form higher order structures through inter-nucleosomal contacts between the histone H4 N-terminal domain (NTD) and the acidic patch of H2A and between two H2B C-terminal domains (CTD) (*Luger et al., 1997*; *Dorigo et al., 2003*, *2004*; *Gordon et al., 2005*; *Schalch et al., 2005*). Histone variants, such as H2A.Z or H2A.Bbd, as well as post-translational modifications of histones, such as H4K16ac, can further regulate the degree of compaction (*Suto et al., 2000*; *Bao et al., 2004*; *Shogren-Knaak et al., 2006*; *Zhou et al., 2007*; *Chandrasekharan et al., 2009*; *Kim et al., 2009*; *Fierz et al., 2011*). Polycomb complexes compact large domains of chromatin (*Eskeland et al., 2010*) and are important for proper development. Histones of the H1 family promote additional compaction by binding between nucleosomes to linker DNA near the DNA entry/exit site on nucleosomes and stabilize the intrinsic ability of nucleosomal arrays to fold in vitro (*Carruthers et al., 1998*; *Robinson et al., 2008*; *Szerlong and Hansen, 2010*). Linker histones may affect chromatin compaction globally (*Fan et al., 2005*), at specific stages of the cell cycle such as mitosis (*Maresca et al., 2005*) or at specific regions of the genome (*Li et al., 2012*). In contrast to canonical histones, the linker histones are much less conserved (*Caterino and Hayes, 2010*), and ectopic expression of human linker histones in the budding yeast even at low levels is lethal for the cell (*Miloshev et al., 1994*). Finally, structural proteins such as condensin also contribute to chromatin condensation (*Tada et al., 2011*). Many of these modulatory mechanisms are dynamic in nature (*Luger et al., 2012*) and may help explain why multicellular organisms can compact chromatin to different degrees in different cell types. However, despite the existence of these mechanisms for genome compaction in higher eukaryotes, it has not been known whether the canonical histones themselves have evolved sequence features that also contribute to the generally increased chromatin compaction observed in organisms with larger genomes.

In this study, we provide evidence from analysis of 160 fully-sequenced eukaryotic genomes that arginine (R) residues at specific positions in the N-terminal tail of histone H2A—which protrudes from the nucleosome on the opposite side of DNA entry/exit site—have co-evolved with increasing genome

size with a concomitant decrease in serines/threonines. Although increases in genome size are associated with phylogenetic evolution from protozoa to fungi to more complex plants and animals, we present genetic and molecular evidence from the budding yeast and human cells as well as in vitro biochemical data to demonstrate that the evolutionary changes in H2A directly regulate chromatin compaction in vivo and in vitro with consequences for the nuclear volume. The evolutionary changes in H2A regulate chromatin compaction in yeast and human cells, revealing a surprising flexibility in the dynamics of the chromatin fiber that has been conserved across distant eukaryotes. This previously unrecognized structural feature of the nucleosome has evolved to enable greater chromatin compaction when genome size is disproportionately larger than the nuclear volume. Our findings also suggest that the reported mutations in the histone H2A NTD may contribute to the altered chromatin compaction that is commonly observed in cancer cells (*Zink et al., 2004*).

## Results

### H2A acquires specifically positioned arginines as genome size increases

To determine whether specific residues in the four core histones have co-evolved with increasing genome size, we performed residue composition analysis of canonical histone protein sequences from 160 fully sequenced eukaryotes with genome sizes ranging from 8 to 5600 Mbp encompassing protozoa, fungi, plants, and animals. The canonical histone proteins for each organism were defined based on at least 90% overlap and 35% identity with the histone fold domain of the corresponding human sequence ('Materials and methods'). Each organism was categorized as having a small (<100 Mbp), medium (100–1000 Mbp), or large (>1000 Mbp) genome (*Figure 1—figure supplement 1A*). Of the canonical histones, the H2A NTD showed the most statistically significant variability in amino acid residues, where the number of arginines increased with increasing genome size (*Figure 1A*), while the number of serines (S) and threonines (T) decreased (*Figure 1B*). Other amino acid residues in the H2A NTD, including lysines (K), did not correlate with genome size (*Figure 1—figure supplement 1B*).

The acquisition of arginines and loss of serines/threonines in the H2A NTD with increasing genome size occur at specific positions and in sequential order. For instance, the human H2A NTD contains arginine residues at positions 3 and 11 that are absent in yeast and at position 20 which is correspondingly a lysine in yeast (*Figure 1C*). In contrast, the human sequence lacks S10 and S15 that are observed in the yeast H2A NTD (*Figure 1C*). Alignment of all H2A NTD sequences also revealed similar trends across all eukaryotes studied here. The heat map in *Figure 1D* shows the occurrence of arginines and serines/threonines in the H2A NTD as a function of genome size (see *Figure 1—source data 1* for raw data and *Figure 1—figure supplement 1C* for statistical analysis). At position 3, an arginine (R3) is predominantly present in medium and large species but is lacking in small species. At position 11, a lysine (K11) is observed in species with medium genomes that evolves to an arginine (R11) mainly in organisms with large genomes. R17 is present in most organisms examined, suggesting a very conserved function for this residue (*Zheng et al., 2010*). At position 20, small genomes contain predominantly a lysine residue, which converts to arginine in medium and large genomes. In contrast, serines/threonines at positions 10 and 15 are found primarily in organisms with small genomes and much less so in organisms with medium and large genomes (*Figure 1D*). Additionally, each of the four H2A NTD arginines is surrounded by a conserved motif (*Figure 1E*). The residues surrounding R3 and R17 are mainly glycine and serine, respectively. At position 11, the motif varies based on genome size. Species with medium-sized genomes contain VKG and those with large genomes contain ARA. The same is true of position 20, where AKA is present in organisms with small genomes and (S/T)RA in larger genome species (*Figure 1E*).

Interestingly, except for R3, the positions of all the other evolutionarily varying residues in the H2A NTD are strongly conserved relative to the histone fold domain and not the N-terminus (*Figure 1F*). When counting conventionally from the N-terminus, amino acids R11, R17, and R20—which are numbered based on the human sequence—were not observed consistently at the same positions in other organisms. However, these residues are respectively 12, 6, and 3 amino acids away from the histone fold in most species (note the vertical axes in *Figure 1F*). S10 and S15—which are numbered based on the yeast sequence—also show more uniform positioning when counted from the histone fold. Altogether, as genome size increases, arginines appear in conserved positions within the H2A NTD relative to the histone fold, and serines and threonines are lost.

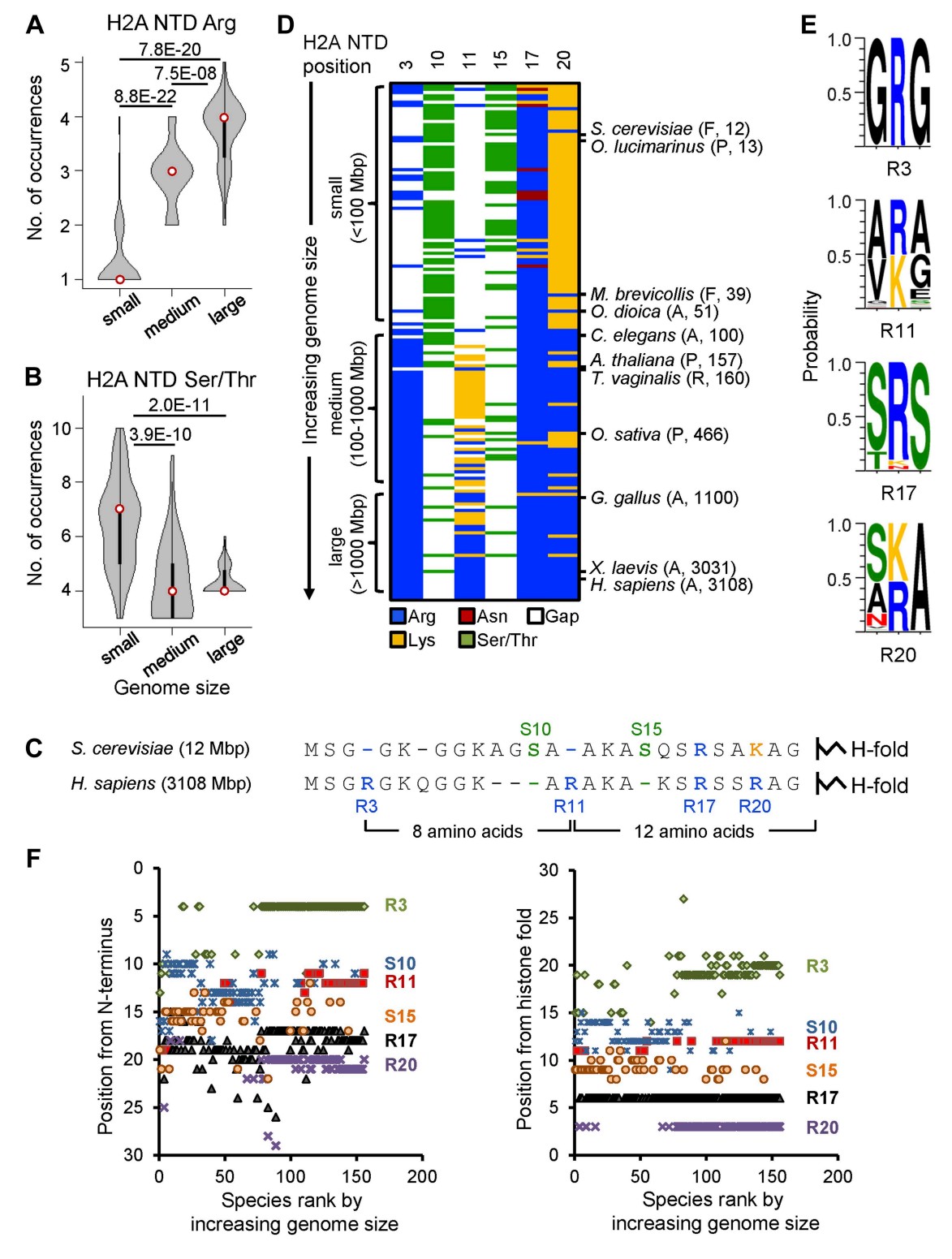

Figure 1. Histone H2A N-terminal sequence has co-evolved with genome size. Violin plots of the number of (A) arginines or (B) serines/threonines in the H2A NTD for species with small, medium, and large genomes. Plot widths correspond to species frequency within each group. (C) H2A NTD sequences for *S. cerevisiae* and *H. sapiens*. (D) Heat map of H2A NTD residue composition at the indicated positions ordered by genome size. Example species are

*Figure 1. Continued on next page*

Figure 1. Continued

shown with kingdom and genome size information. (**E**) Protein sequence motifs surrounding the four H2A NTD arginine residues. (**F**) Positioning of evolutionarily variable residues relative to the H2A N-terminus (left) or histone fold (right). See also *Figure 1—figure supplements 1 and 2*.

The following source data and figure supplements are available for figure 1:

**Source data 1**. H2A multiple sequence alignments, heat map data, and canonical H2A isoforms.

**Figure supplement 1**. Phylogenetic distribution of species analyzed in this paper.

**Figure supplement 2**. H2A arginines 3 and 11 are situated adjacent to DNA within the nucleosome.

## Arginines 3 and 11 in the H2A NTD increase chromatin compaction

To determine whether arginines and serines/threonines of the H2A NTD affect chromatin compaction in vivo, we took advantage of a strain of *Saccharomyces cerevisiae*, that has both chromosomal copies of H2A deleted and carries a single copy of H2A on a plasmid (TSY107), to construct mutant strains containing single or multiple insertions of arginines into their conserved motifs, deletions of serines, or combinations thereof (see *Table 1* for specific amino acid changes and *Supplementary file 1A* for a description of the mutant strains). Two mutants, R3(ΔGS10)R11 and R11ΔS15, were also designed such that the spacing between R3 and R11 or R11 and the histone fold, respectively, is the same as in the H2A NTD of organisms with large genomes (*Figure 1C,F*). As a control for positive charge, mutant strains with lysines inserted in the same positions as arginines were also generated.

To test the effects of H2A NTD changes on chromatin compaction, the physical distance between two probes on chromosome XVI spaced 275 kb apart was assessed in each of the H2A mutants using fluorescent in situ hybridization (FISH) (*Figure 2A*; *Guacci et al., 1997*; *Bystricky et al., 2004*). The probes were differentially labeled and visualized by confocal microscopy. The distance between the probes was measured in a single plane in which both probes were present within each nucleus (*Bystricky et al., 2004*). When compared to the isogenic wild type (WT), addition of a single arginine at position 3 (R3) or 11 (R11) to the H2A NTD was sufficient to significantly decrease the average inter-probe distance by 18% and 15%, respectively (*Figure 2B,C*; *Table 1*). The average interprobe distance was further decreased by 22% when both arginines were present (R3R11) and even more so (30%) in R3(ΔGS10)R11. Deleting G9S10 (ΔGS10) alone caused slightly increased compaction with low statistical significance (*Table 1*). The largest decrease in interprobe distance (41%) was observed in the R11ΔS15 mutant, which places R11 12 amino acids from the histone fold, the same position as in organisms with large genomes. Removal of S15 (ΔS15) alone had no effect. The effect of arginines was not simply due to increasing the positive charge of the H2A NTD, as insertions of lysines at positions 3 and 11 did not significantly affect the interprobe distances (*Figure 2C* and *Supplementary file 2*). Although lysines are found at these positions in certain species (*Figure 1D*), the lack of potential compaction by lysines may be due to the absence of other evolutionary changes in yeast histones (see *Figure 1E*). Additionally, R17K or K20R mutations did not affect compaction, nor did a randomly inserted arginine at position 6 (R6) (*Figure 2C* and *Supplementary file 2*), suggesting that not every arginine in the H2A NTD contributes to chromatin compaction.

We further confirmed the effects of R11 on chromatin compaction using three additional probe sets (*Figure 2A*). The level of compaction seen in our WT strain is similar to what has been previously reported in yeast using a different strain background (*Bystricky et al., 2004*). The interprobe distances for all probe sets were significantly decreased in R11 compared to WT and even more so in R11ΔS15 but not ΔS15 alone (*Figure 2—figure supplement 1A–C* and *Supplementary file 2*). Plotting the physical vs genomic distances for all probe sets revealed uniform compaction across large genomic distances (*Figure 2D*). The effect of R11 on chromatin compaction was not strain-specific as H2A R11 and R11ΔS15, but not ΔS15, caused chromatin compaction in a different strain background (*Figure 2E*; *Supplementary file 2*). We therefore conclude that H2A arginines at positions 3 and 11, especially when R11 is placed at the evolutionarily-conserved position relative to the histone fold, increase the degree of chromatin compaction.

Chromatin is differentially compacted at different cell cycle stages (*Guacci et al., 1997*). Cell cycle profile analysis showed little difference between the strain harboring WT H2A and any of the mutant

**Table 1.** List of H2A mutations, sequence changes and their effects on chromatin compaction and nuclear volume

| H2A mutant | H2A NTD Protein sequence | FISH | | Nuclear volume | |
|---|---|---|---|---|---|
| | | % Change | p-value | % Change | p-value |
| Yeast | | | | | |
| WT | SG–GKG–GKAGSA–AKASQSRSAKAG | – | 1.0E+00 | – | 1.0E+00 |
| R3 | SG**R**GKG–GKAGSA–AKASQSRSAKAG | **−18** | 9.5E−04 | −5 | 4.1E−01 |
| R11 | SG–GKG–GKAGSA**R**AKASQSRSAKAG | **−15** | 8.6E−04 | **−20** | 5.9E−05 |
| R3R11 | SG**R**GKG–GKAGSA**R**AKASQSRSAKAG | **−22** | 8.2E−06 | **−16** | 3.0E−03 |
| R3(ΔGS10)R11 | SG**R**GKG–GKA · · A**R**AKASQSRSAKAG | **−30** | 2.1E−06 | +6 | 3.7E−01 |
| R11ΔS15 | SG–GKG–GKAGSA**R**AKA · QSRSAKAG | **−41** | 3.9E−08 | **−9*** | 4.7E−04 |
| K3 | SG**K**GKG–GKAGSA–AKASQSRSAKAG | +9 | 8.6E−01 | **+13** | 9.4E−03 |
| K11 | SG–GKG–GKAGSA**K**AKASQSRSAKAG | +16 | 3.1E−01 | +3 | 2.6E−01 |
| K3K11 | SG**K**GKG–GKAGSA**K**AKASQSRSAKAG | +6 | 8.3E−01 | **+31** | 5.4E−08 |
| K11ΔS15 | SG–GKG–GKAGSA**K**AKA · QSRSAKAG | −7 | 9.2E−02 | +2 | 6.6E−01 |
| ΔGS10 | SG–GKG–GKA · · A–AKASQSRSAKAG | −6 | 3.2E−02 | +10 | 3.0E−02 |
| ΔS15 | SG–GKG–GKAGSA–AKA · QSRSAKAG | +3 | 9.4E−02 | **+9** | 9.7E−03 |
| R6 | SG–GKG**R**GKAGSA–AKASQSRSAKAG | −5 | 5.6E−02 | **+10** | 1.0E−03 |
| K20R | SG–GKG–GKAGSA–AKASQSRSA**R**AG | −3 | 3.0E−01 | +7 | 1.5E−02 |
| R17K | SG–GKG–GKAGSA–AKASQS**K**SAKAG | −1 | 7.9E−01 | 0 | 2.7E−01 |
| Human—HA Tag | | | | | |
| WT | SGRGKQGGKTRAKAKSRSSRAG | – | 1.0E+00 | – | 1.0E+00 |
| ΔR3 | SG · GKQGGKTRAKAKSRSSRAG | **+39** | 8.3E−03 | **+42** | 1.3E−08 |
| R11K | SGRGKQGGKT**K**AKAKSRSSRAG | **+20** | 2.3E−02 | **+14** | 1.2E−03 |
| R11A | SGRGKQGGKT**A**AKAKSRSSRAG | **+43** | 1.0E−05 | **+21** | 5.7E−07 |
| ΔR3R11A | SG · GKQGGKT**A**AKAKSRSSRAG | **+35** | 3.5E−03 | **+18** | 5.9E−04 |
| Human—FLAG Tag | | | | | |
| WT | SGRGKQGGKARAKAKSRSSRAG | – | 1.0E+00 | – | 1.0E+00 |
| Δ1–12 | · · · · · · · · · · · · KAKSRSSRAG | **+47** | 4.9E−03 | **+18** | 2.8E−04 |

The -marks indicate spacing for sequence alignment purposes. The inserted residues are bold typed and underlined. Deletions are indicated by ·.

Percent (%) change refers to the difference in median values relative to WT unless otherwise indicated; the statistically significant differences are bold typed. p-values were calculated using the t-test (yeast) and Mann–Whitney U test (human).

*percent change was calculated relative to isogenic WT control (ΔS15).

strains (*Figure 2—figure supplement 1D–E*), indicating that the observed differences in chromatin compaction are not due to altered cell cycle profiles. Chromatin compaction may also be influenced by nucleosomal spacing; indeed the linker DNA length is larger in human cells than in yeast (*Grigoryev, 2012*). We find that there are essentially no differences in nucleosomal density in H2A arginine mutants using Micrococcal nuclease (MNase) digestion (*Figure 2F*, *Figure 2—figure supplement 1F*), indicating that the average nucleosomal spacing is not affected by these mutations. But the more compact mutants displayed decreased accessibility to MNase as indicated by the delayed appearance of the nucleosomal digestion pattern (*Figure 2F*, *Figure 2—figure supplement 1F*).

## H2A arginines and serines affect nuclear volume in yeast

Since chromatin structure may influence the volume of the nucleus (*Cavalier-Smith, 2005*), we asked whether nuclear volume was affected by H2A-mediated chromatin compaction. We tagged a nuclear pore protein, Nup49, in its chromosomal locus with GFP to visualize the nuclear membrane and used confocal microscopy to capture three-dimensional images of the nucleus to quantify volumes

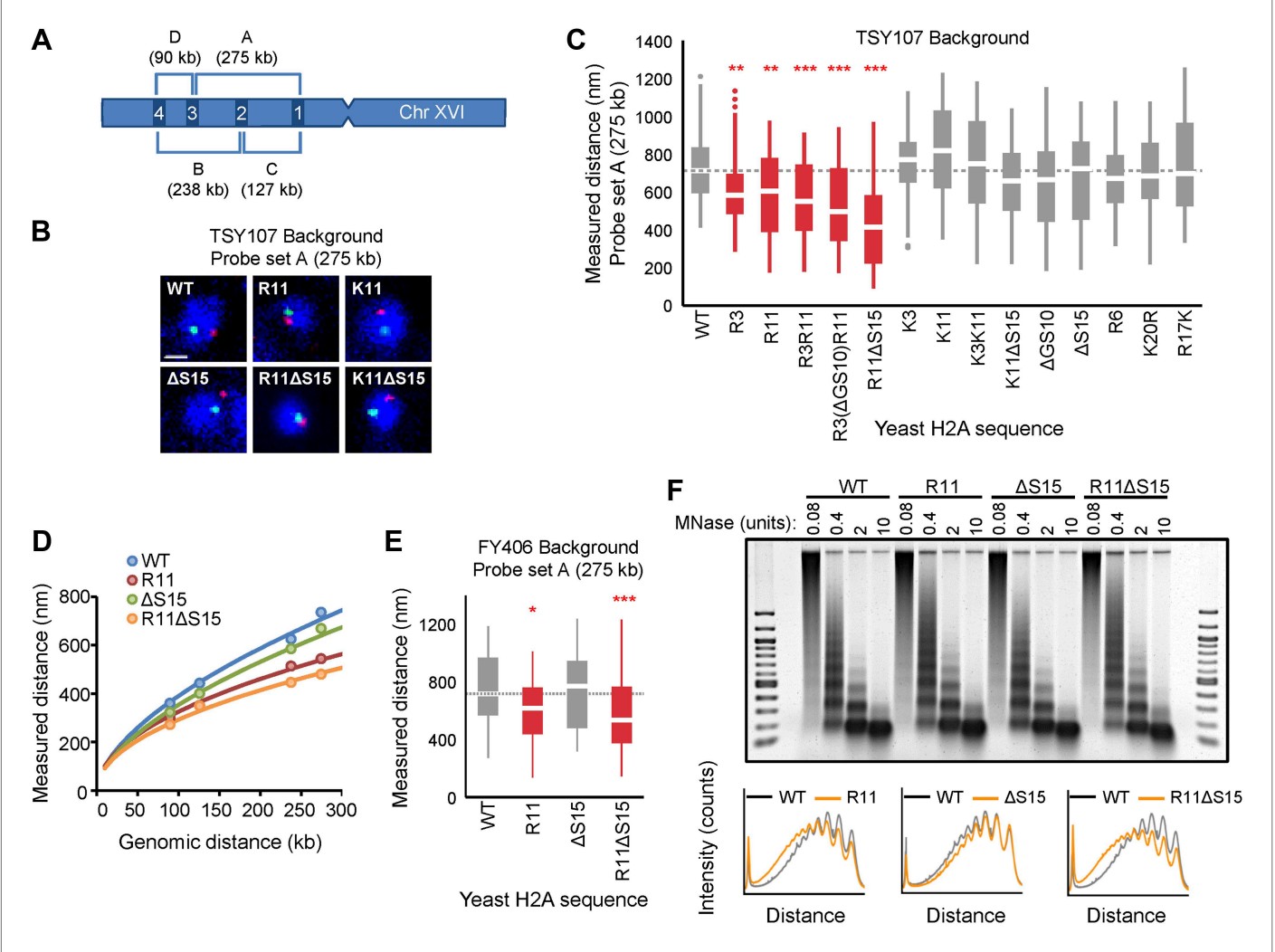

**Figure 2**. Ectopic expression of H2A NTD arginines causes compaction in yeast. (**A**) Schematic position of probes on chromosome XVI that were used for FISH. The letters correspond to the probe sets. (**B**) FISH images and (**C**) boxplot of the distributions of interprobe distances for probe set A in the indicated strains. (**D**) The mean interprobe distances for the indicated yeast strains for probe sets A, B, C, and D are plotted as a function of genomic distance. Solid lines are best fit equations. (**E**) Boxplot of the distributions of interprobe distances for probe set A in the indicated strains. Dashed lines mark the median value for the WT strain. The boxplot whiskers contain 90% of the data. All scale bars are 1 μm. Boxes are colored if the mean of the indicated strain is significantly different from WT. Red stars denote level of significance: *p<0.01; **p<0.001; ***p<0.0001 (For exact values, see *Supplementary file 2*). (**F**) Agarose gel electrophoresis of MNase-digested chromatin in the indicated strains including the densitometric profiles comparing the WT to each of the mutant H2A strains for a given amount of enzyme. See also *Figure 2—figure supplement 1*.

The following figure supplements are available for figure 2:

**Figure supplement 1**. Ectopic expression of H2A NTD arginines causes compaction in yeast.

of ≥150 cells per H2A mutant (*Figure 3A,B*; *Table 1*; *Supplementary file 3*, see 'Materials and methods' for volume calculations). As compared to WT cells, H2A mutants containing R11 or R3R11, both of which contain more compact chromatin, displayed significantly decreased nuclear volumes. The average nuclear volume in the R3 mutant was also less than WT but did not reach statistical significance. Interestingly, H2A mutants from which serines 10 and 15 were removed displayed larger nuclear volumes. Simultaneous insertions of arginines into these strains (R3(ΔGS10)R11 and R11ΔS15) decreased their nuclear volume (R11ΔS15 p<0.001 compared to ΔS15), restoring them to levels similar to WT. The control strains with either lysines or R6 had nuclear volumes similar to or larger than WT. Neither arginines nor serines had any effect on total cell size as measured by concanavalin A staining

(*Figure 3—figure supplement 1A,B*; *Supplementary file 3*). In the FY406 strain background, ΔS15 did not cause an increase in nuclear volume; and thus both R11 and R11ΔS15 strains exhibited smaller nuclear volumes than isogenic WT (*Figure 3C*; *Supplementary file 3*). These data suggest that modulation of chromatin compaction through the H2A NTD, especially in the presence of R11, affects the nuclear volume but this effect may be indirect (see human data below).

## Loss of H2A arginines causes de-compaction of chromatin in human cells

Since the H2A NTD in large genomes contains both R3 and R11, we expected that their removal would cause de-compaction of chromatin. To test this prediction, we ectopically expressed WT or mutant H2A in several human cell lines and measured the distances between probes 0.49 Mbp apart on chromosome 1 by FISH, as well as the largest nuclear cross-sectional areas ('Materials and methods'). The H2A gene was HA-tagged and mutated to remove R3 (ΔR3), to replace R11 with alanine (R11A) or lysine (R11K), or to combine two mutations (ΔR3R11A). The H2A constructs were overexpressed using the strong CMV promoter in the normal human IMR90 fibroblasts, the breast cancer cell line MDA-MB-453, or the HEK293 cells. Cells overexpressing ΔR3, R11A, or ΔR3R11A H2A mutants had increased interprobe distances, indicating de-compaction of chromatin. Expression of H2A R11K had modest effects on chromatin de-compaction with marginal statistical significance (*Figure 4A,B*, *Figure 4—figure supplement 1A,B*; *Supplementary file 4*). Cells expressing any of the H2A mutants displayed larger nuclear areas, suggesting that nuclear size is increased (*Figure 4C,D*, *Figure 4—figure supplement 1C–F*; *Supplementary file 5*). Equal degrees of overexpression were confirmed by immunofluorescence analysis with an anti-HA antibody and detection of HA-H2A by western blotting (*Figure 4D*, *Figure 4—figure supplement 1G*). Ectopic expression of a C-terminally FLAG-tagged H2A mutant missing residues 1–12 (Δ1–12) also caused significant de-compaction of chromatin and increased nuclear area despite being expressed at a lower level than WT (*Figure 4E,F*). These data

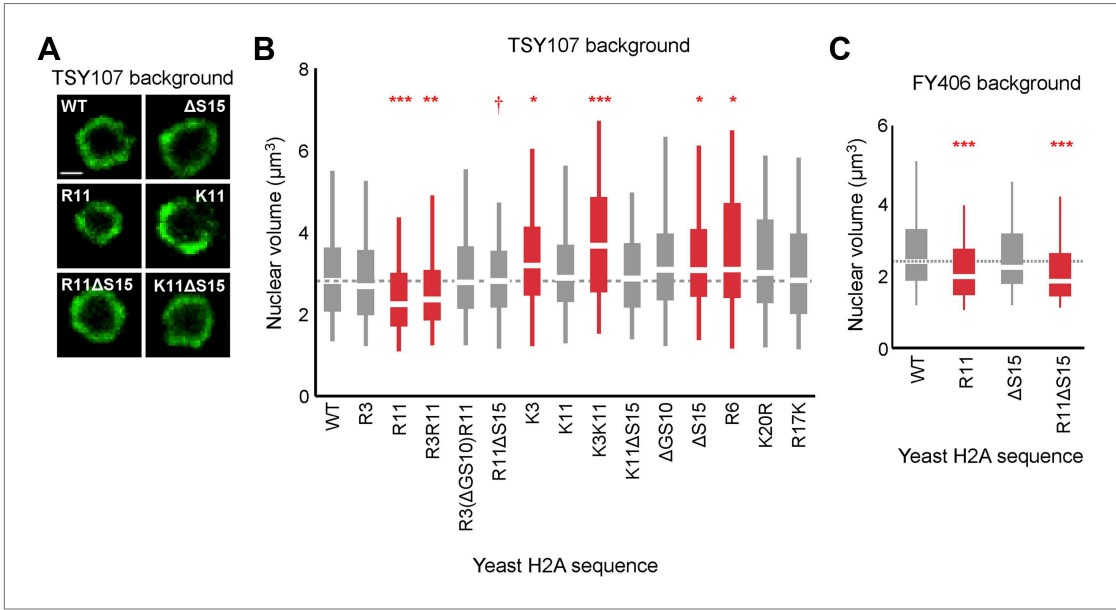

**Figure 3**. Ectopic expression of H2A NTD arginines decreases nuclear volume in yeast. (**A**) Images of the nuclear envelope, as visualized by Nup49p-GFP, and boxplot of the distributions of nuclear volumes in the indicated strains in the TSY107 background (**B**) or the FY406 background (**C**). Dashed lines mark the median value for the WT strain. The boxplot whiskers contain 90% of the data. All scale bars are 1 µm. Boxes are colored if the mean of the indicated strain is significantly different from WT. Red stars denote level of significance: *p<0.01; **p<0.001; ***p<0.0001 (*Supplementary file 3*). Red dagger (†) indicates that mean nuclear volume of R11ΔS15 is significantly smaller than its isogenic WT strain (ΔS15; p<0.001). See also *Figure 3— figure supplement 1*.

The following figure supplements are available for figure 3:

**Figure supplement 1**. H2A arginines do not affect cell size.

demonstrate that, consistent with our predictions, the H2A NTD, especially arginines 3 and 11, function to compact chromatin in human cells.

## H2A R11 regulates compaction of nucleosomal arrays in vitro

Because R11 compacts chromatin in vivo, we investigated whether this effect is directly on the chromatin fiber. We used step-wise salt dialysis to assemble nucleosomal arrays with a DNA template containing 12 copies of the 177 bp '601' nucleosome positioning sequence (601-177-12) and recombinant *Xenopus laevis* histone octamers that contain either WT H2A or one with R11 deleted (ΔR11). We assembled nucleosomal arrays at different octamer-to-template ratios (0.9, 1, and 1.1 octamer to 1 template) and monitored the quality of the arrays by $MgCl_2$ precipitation and restriction digest analysis using ScaI. We found that a 1:1 octamer-to-template ratio gave the best results as the ScaI digest demonstrated well-assembled arrays compared to the 5% free DNA loaded as a comparison (*Figure 5A*). We used analytical ultracentrifugation to determine the sedimentation velocity combined with van Holde–Weischet analysis (*Weischet et al., 1978*) to ascertain the distribution of sedimentation coefficients (S) for each nucleosomal array in the absence or presence of 0.8 mM $MgCl_2$, a concentration of the divalent cation that promotes intra-molecular folding of nucleosomal arrays (*Schwarz and Hansen, 1994*). In the absence of $Mg^{2+}$, arrays containing WT H2A sedimented with a coefficient of 33.1, which is a value that has been previously shown for similar arrays (*Dorigo et al., 2003*; *Shogren-Knaak et al., 2006*; *Zhou et al., 2007*). In contrast, arrays missing R11 adopted a more extended conformation with a smaller sedimentation coefficient of 31.0 (*Figure 5B*). Addition of $Mg^{2+}$ increased compaction of both arrays and shifted the sedimentation coefficients to 39.3 and 37.4 for WT and ΔR11 H2A, respectively (*Figure 5B*). A second independent chromatin assembly and ultracentrifuge analysis confirmed these results (*Figure 5—figure supplement 1*). Thus, in the absence of R11 in the H2A NTD, nucleosomal arrays adopt a less compact conformation even in the presence of divalent cations, showing that R11 directly increases chromatin compaction.

## Compaction of chromatin by H2A NTD arginines does not alter global gene expression in yeast

To determine whether chromatin compaction through H2A arginines interferes with transcription regulation, we examined gene expression patterns in the H2A yeast mutants. Remarkably, there was a high level of correlation (≥0.99) between all strains examined (*Figure 6A*), and no specific gene ontology was found among the genes that were differentially expressed by twofold or more. The expression levels of the histone genes were similar, indicating that altered levels of histone genes expression do not account for the changes in chromatin compaction. These data indicate that compaction of chromatin by H2A does not significantly alter global gene expression in exponentially growing cells.

All strains also showed similar growth rates in rich media (*Figure 6B*) and no significant differences in sensitivity to hydroxyurea, methyl methanesulfonate (MMS), bleomycin, 4-nitroquinoline 1-oxide (4NQO), cycloheximide, and rapamycin, indicating no major defects with DNA replication or repair, protein synthesis, or the TOR signaling pathways (*Figure 6C*). But in competition growth assays in which equal amounts of WT and H2A mutant cells harboring the PGK1 gene fused to either GFP or RFP were co-cultured, the H2A mutants regardless of any effect on chromatin compaction, were outcompeted (*Figure 6D*). This suggests that changes in the H2A NTD sequence can affect the overall fitness of the cell.

## H2A NTD arginines and their surrounding residues are mutated in cancer and affect chromatin compaction

Deregulated chromatin compaction is often a pathological hallmark of cancer cells (*Edens et al., 2012*), although the underlying mechanisms are not well-understood. A survey of the COSMIC database (*Forbes et al., 2011*), as of the time of writing, revealed 41 documented missense mutations within the H2A NTD with 29 (71%) affecting a residue within one of the four arginine motifs (*Figure 7A*). R11, which had the strongest effect of any single arginine residue on chromatin compaction, is the most commonly mutated residue in the H2A NTD. We tested the effects of three of these mutations, R11C, H, and P and found that ectopic expression of each in normal human fibroblasts decreases chromatin compaction significantly with R11P having the strongest effect (*Figure 7B,C*). These cancer mutations have little effect on increasing nuclear area, however (*Figure 7D,E*), in contrast to R11A (*Figure 4D*). It is unclear to what extent the H2A mutants have to be expressed in cancer cells relative to the 17 canonical H2A genes in the human genome to affect chromatin compaction. But our data

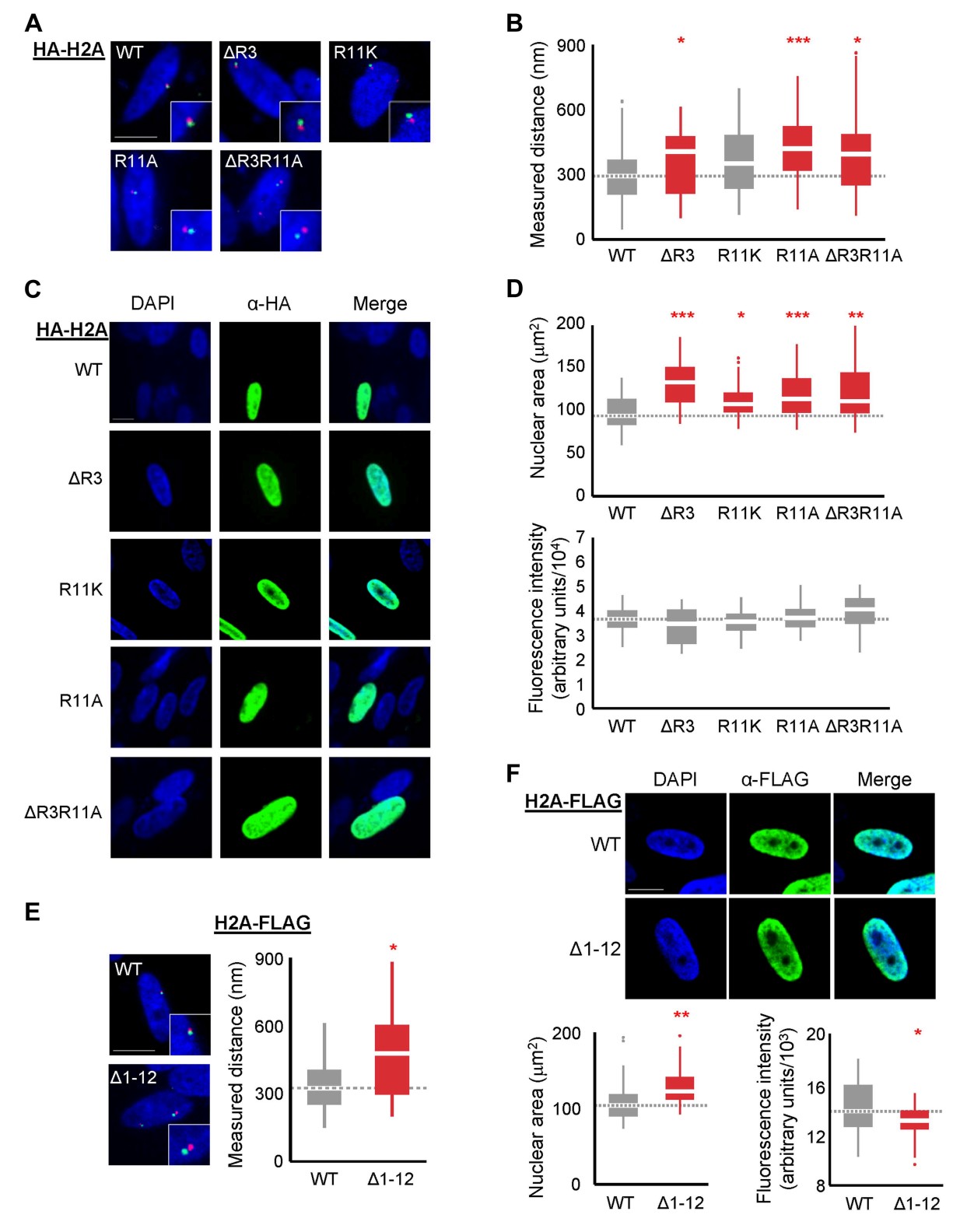

**Figure 4**. Loss of H2A NTD arginines decreases chromatin compaction in human cells. (**A**) FISH images of probes on chromosome 1 in normal primary IMR90 fibroblasts with HA-tagged WT or mutant H2A overexpressed as indicated. (**B**) Boxplot of the distributions of inter-probe distances. Note that R11K was only marginally significant at p=0.023. (**C**) Immunofluorescence images of IMR90 cells overexpressing HA-tagged WT or mutant H2A as
*Figure 4. Continued on next page*

*Figure 4. Continued*

indicated. (**D**) Top: boxplot of the distributions of largest nuclear cross-sectional areas in the indicated H2A overexpressing cells. Bottom: boxplot of the distributions of α-HA fluorescence intensities. (**E**) Left: FISH images, as in (**A**), of IMR90 cells expressing a C-terminal FLAG-tagged WT or tailless (Δ1–12) H2A. Right: boxplot of the distributions of inter-probe distances. (**F**) Top: immunofluorescence images of IMR90 cells overexpressing FLAG-tagged WT or tailless H2A. Bottom: boxplot of nuclear areas and fluorescence intensities, as indicated. Dashed lines mark the median value for the WT strain. All scale bars are 10 μm. Boxes are colored if the mean of the indicated strain is significantly different from WT. Red stars denote level of significance: *p<0.01; **p<0.001; ***p<0.0001 (**Supplementary files 4 and 5**). See also **Figure 4—figure supplement 1**.

The following figure supplements are available for figure 4:

**Figure supplement 1**. Loss of H2A NTD arginines decreases chromatin compaction in human cells.

suggest that over-expression of an H2A mutant has the potential to disrupt chromatin compaction in cancer.

## Discussion

In this study, we describe evolutionary adaptations of the histone H2A whereby single arginines in the NTD function to dramatically affect the degree of genome compaction. This mechanism is distinct from several other known chromatin compaction mechanisms in higher eukaryotes (**Bednar et al., 1998**; **Dorigo et al., 2003**; **Shogren-Knaak et al., 2006**; **Zhou et al., 2007**; **Fierz et al., 2011**) in that it involves the histone proteins themselves. The H2A-mediated chromatin compaction thus provides a novel but potentially complementary mechanism for genome compaction. Organisms with small genomes but also very small cell size may face similar physical constraints as those with larger genomes, and may therefore use arginine-containing H2A as a means for chromatin compaction. For instance, *Ostreococcus tauri* which possesses an R3-containing H2A, is a free-living unicellular algae that has a very small genome of 12.6 Mbp but a cell diameter of 0.8 μm (**Palenik et al., 2007**). *S. cerevisiae*, which does not contain an H2A with R3, has a similarly sized genome but has a cell diameter that is approximately five times larger. Furthermore, certain organisms such as *Oikopleura dioica*, which has one of the smallest genomes in animals, have distinctive life cycles and possess H2A genes with and

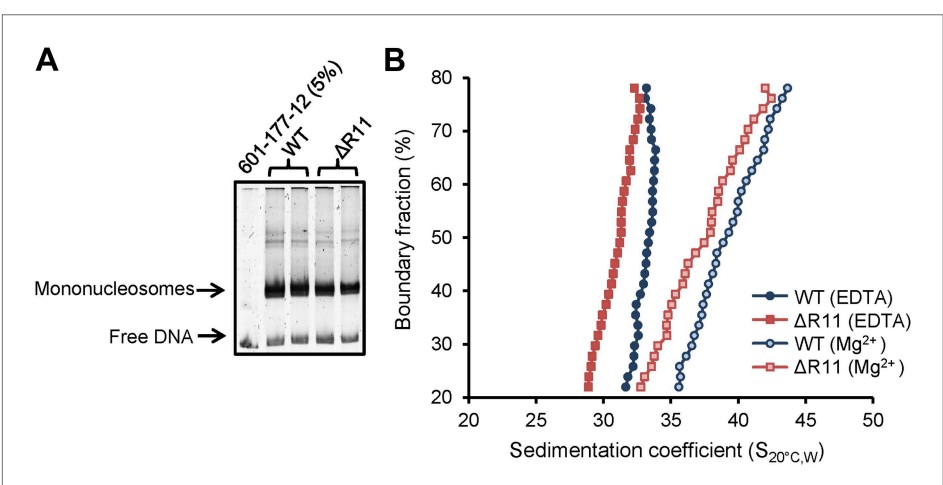

**Figure 5**. H2A NTD R11 directly modulates the compaction of chromatin fibers in vitro. (**A**) Polyacrylamide gel electrophoresis (PAGE) of ScaI-digested 601-177-12 DNA template assembled with octamers containing recombinant WT or ΔR11 H2A. As a control, 5% of the 601-177-12 DNA without octamers was also digested. (**B**) The distribution of sedimentation coefficients determined by van Holde-Weischet analysis plotted against the percent boundary fraction in the absence or presence of 0.8 mM MgCl₂ as indicated. $S_{20°C,W}$ is the sedimentation coefficient corrected to water at 20°C. See also **Figure 5—figure supplement 1**.

The following figure supplements are available for figure 5:

**Figure supplement 1**. H2A NTD R11 directly modulates the compaction of chromatin fibers in vitro.

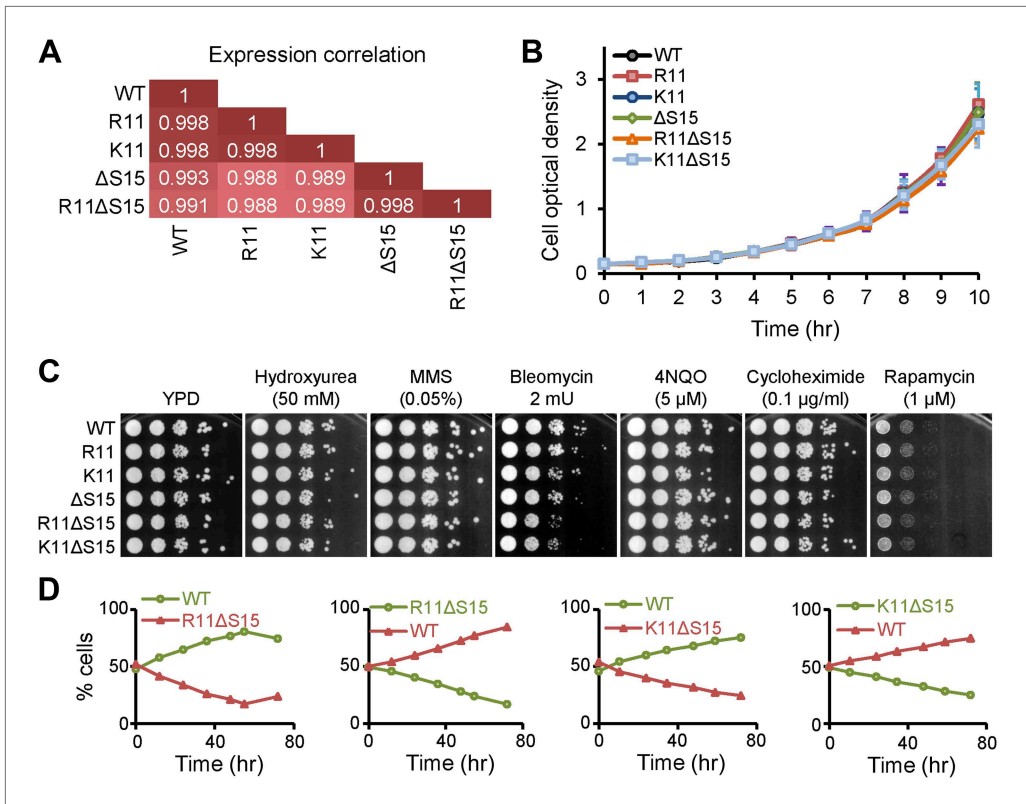

**Figure 6**. Mutations to H2A NTD decrease the fitness of yeast. (**A**) Pearson correlations between the global gene expressions of the indicated strains grown in YPD. Correlations are calculated from an average of at least two experiments. (**B**) Growth curves of the indicated H2A yeast strains over 10 hr in YPD. (**C**) Spot tests with 10-fold serial dilutions for the indicated strains in the presence of different drugs. (**D**) The proportion of yeast cells in a co-culture of WT and the indicated mutant H2A carrying Pgk1 gene fusion to GFP (green) or RFP (red) as indicated by color.

without arginines, which may enable them to dynamically regulate genome compaction at different stages of their life cycles (**Moosmann et al., 2011**) (for species with H2A isoforms, see attached spreadsheet). So, the H2A arginines may have evolved in circumstances when the genome size became disproportionately large compared to nuclear volume. Interestingly, the toad, *Bufo gargarizans*, which has a genome size that is twice as large as the human genome, possesses an H2A gene with not only R3 and R11 but also glutamine 6 replaced with an arginine, suggesting that additional arginines in the H2A tail may enable further compaction in organisms with even larger genomes.

To better understand the three dimensional positions of the H2A NTD arginines, we examined a crystal structure of the mono-nucleosome in which R3, R11, R17, and R20 were all simultaneously crystalized, and visualized interactions between nucleosomes within the crystal lattice (**Davey et al., 2002**). Interestingly, while R17 and R20 are more buried within the octamer, R3 and R11 are situated close to the DNA backbone. R3 is at 2.87 Å from the DNA and could potentially bind the DNA gyre as the DNA wraps around the histone octamer. R11 forms close contacts with the DNA phosphate backbone of self and neighboring nucleosomes (4.09, 2.90 Å, respectively). Although these interactions may have helped form the crystal lattice, they also suggest a possible mechanism for tighter nucleosomal stacking in vivo through shielding of the DNA negative charge (**Figure 1—figure supplement 2A,B**; **Davey et al., 2002**). Thus, the evolutionary appearance of arginines in the H2A NTD sequence at positions 3 and 11 corresponds to strategic positioning of R3 and R11 within the nucleosome structure that may enable interactions with the DNA, leading to more compact chromatin.

Our in vivo data in yeast cells demonstrate that interprobe distances shorten in the presence of the H2A arginines R3 and R11. While the mechanism of this shortening is still unknown, the two most likely explanations are due to linear chromosomal compaction or increased chromatin looping

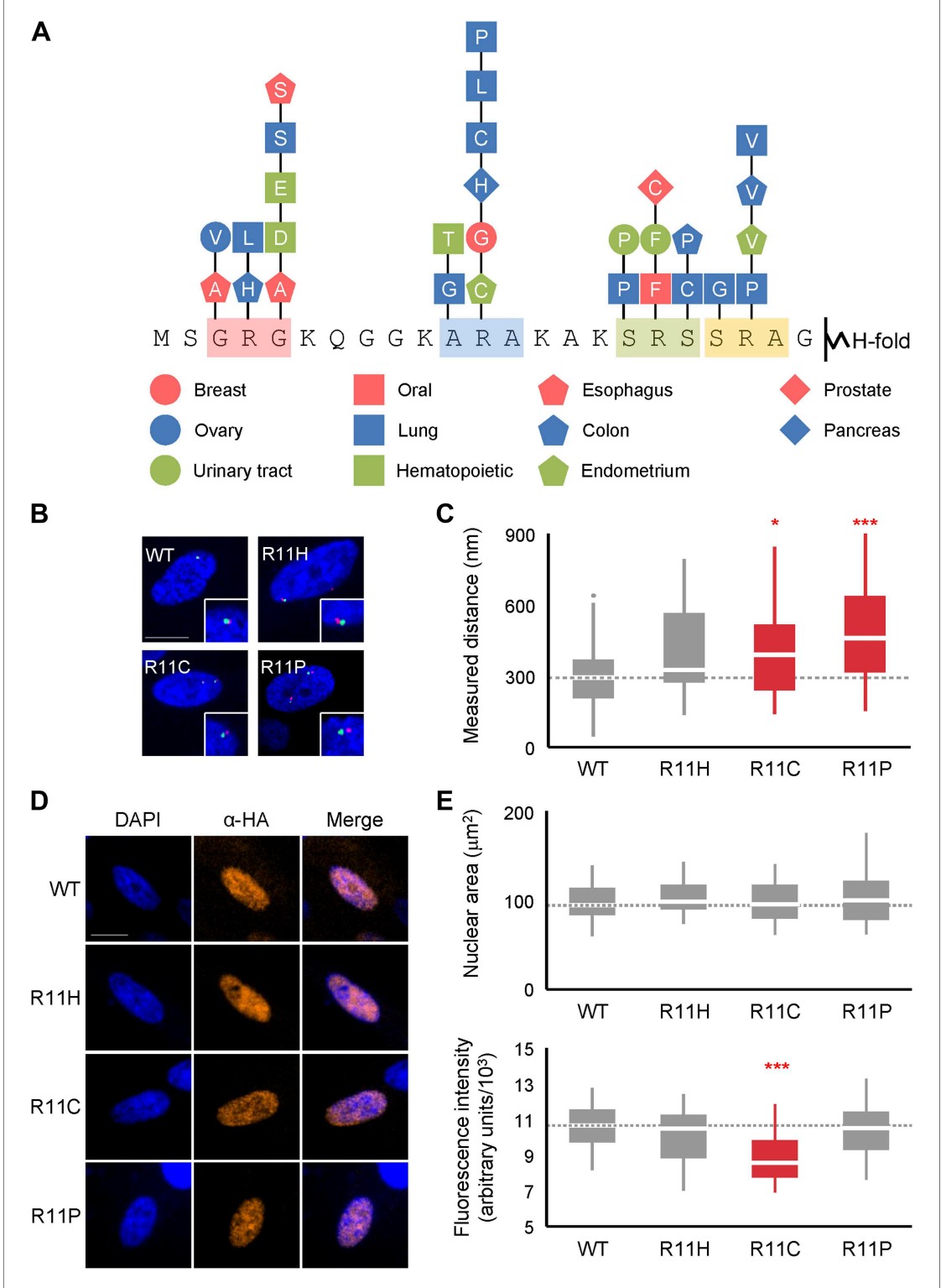

**Figure 7**. Mutations of H2A NTD found in cancers decreases chromatin compaction in human cells. (**A**) Schematic of the H2A NTD showing only the mutations within the arginine motifs found in various cancers as indicated by the colored shapes (**Forbes et al., 2011**). The letter within each shape represents the mutated amino acid. (**B**) FISH images of probes on chromosome 1 in normal primary IMR90 fibroblasts with HA-tagged WT or mutant H2A overexpressed as indicated. (**C**) Boxplot of the distributions of inter-probe distances. (**D**) Immunofluorescence images of IMR90 cells overexpressing HA-tagged WT or mutant H2A as indicated. Anti-HA primary and Alexa Fluor 647-conjugated secondary antibodies were used to determine expression
*Figure 7. Continued on next page*

*Figure 7. Continued*

in FISH images and for measurement of nuclear areas. (**E**) Top: boxplot of the distributions of largest nuclear cross-sectional areas in the indicated H2A overexpressing cells. Bottom: boxplot of the distributions of α-HA fluorescence intensities. Dashed lines mark the median value for the WT strain. All scale bars are 10 µm. Boxes are colored if the mean of the indicated strain is significantly different from WT. Red stars denote level of significance: *p<0.01; ***p<0.0001 (*Supplementary files 4 and 5*).

(*Bohn and Heermann, 2010*). However, our data are more consistent with increased linear compaction due to several reasons. First, our analysis of multiple probes along Chr XVI in yeast demonstrates a uniform compaction between all probe pairs examined. Second, our in vivo data with human cell lines shows de-compaction of chromatin in the absence of R11. If chromatin looping was the mechanism, loops would have to be disassembled in human nuclei independent of factors such as CTCF and condensin. Third, our in vitro data show that R11 alone affects chromatin compaction even in absence of divalent cations. Because the in vitro experiments were performed with unmodified histones in arrays with equal linker lengths, this strongly points to a direct effect of H2A NTD arginines on chromatin compaction that occurs in short arrays. Both R3 and R11 are at contact distances from the DNA, and R11 may also contact the DNA backbone of the neighboring nucleosome (*Figure 1—figure supplement 2*). These intra- and inter-nucleosomal interactions with the arginines and the DNA may serve to neutralize the negative charge of the DNA backbone, leading to enhanced stacking of nucleosomes and hence increased compaction. Consistent with this model, the other two arginine residues in the H2A NTD, R17 and R20, which are more buried from the surface, do not affect compaction by themselves. However, their functions may be to modulate the effects of the surrounding residues. Although all canonical H2A genes contain R3 and R11 in humans, the cell may still be able to dynamically regulate chromatin compaction by these arginines. For instance, arginines may be subject to posttranslational modifications, such a methylation which makes the arginine residue bulkier, or citrullination which removes the positive charge (*Wang et al., 2001*; *Hagiwara et al., 2005*; *Di Lorenzo and Bedford, 2011*; *Waldmann et al., 2011*). Interestingly, the H2A NTD is situated in close proximity to the H2B CTD which when ubiquitylated, disrupts chromatin compaction in vitro (*Fierz et al., 2011*), lending support to the ability of this region of the nucleosome to modulate chromatin compaction.

The inability of lysines, especially at position 11, to increase chromatin compaction suggests exquisite structural constraints for H2A-mediated chromatin compaction. Although lysines and arginines both are positively charged, the positive charge of arginine is due to the presence of a guanidinium group that is structurally different from the positive charge of an amino group of a lysine residue. In this regard, it is interesting to note that only arginines contact DNA as it wraps around the nucleosome core (*Luger et al., 1997*); and arginines preferentially bind the minor groove of DNA compared to lysines (*Rohs et al., 2009*). The context in which lysines appear in evolution may be important as well. We did not observe a lysine at position 3 in our list of organisms, and K11 was present in organisms with medium-sized genomes and surrounded mainly by the motif VKG (*Figure 1D,E*). When tested in our *S. cerevisiae* strains, K11 was in the context of AKA. So, it is conceivable that additional amino acid changes would be required for lysines in the H2A NTD to increase genome compaction.

The nucleoskeletal theory proposes that chromatin structure influences the shape of the nucleus, and thus is a major determinant of nuclear volume (*Cavalier-Smith, 2005*), although the amount of DNA per se does not affect nuclear volume (*Neumann and Nurse, 2007*). Non-chromatin components such as nuclear import factors from the cytoplasm may also modulate nuclear size (*Levy and Heald, 2010*). Our data suggest that in particular cases, the effects of H2A NTD mutations on chromatin compaction are linked to nuclear volume, although not in a straightforward relationship. While arginines at positions 3 and 11 increase chromatin compaction and reduce nuclear volume, lysines at the same positions have no effect on chromatin compaction yet increase nuclear volume. Removal of serines at positions 10 or 15 has little effect on compaction but also increase nuclear volume. In human cells, expression of all R11 mutants (R11A, C, H, and P) decreased compaction but only R11A also affected nuclear area. Although we do not observe a clear-cut relationship between nuclear volume and chromatin compaction, our data identify a region of the nucleosome that is directly or indirectly linked to nuclear volume control mechanisms.

Since alterations in chromatin structure often cause changes in transcription (*Parra and Wyrick, 2007*), we were surprised that mutant H2A-containing yeast had very similar gene expression profiles as WT cells, grew at similar rates, did not have altered cell cycle profiles, and were not sensitive to DNA

damaging drugs or environmental challenges. These data raise the possibility that H2A-mediated compaction of chromatin may have evolved as a mechanism to enable regulation of chromatin compaction without having to make compensatory changes to all other processes that are also based on DNA such as transcription. Nevertheless, it remains to be determined what molecular or cellular processes govern the optimal level of chromatin compaction and nuclear volume for an organism. The stable alterations of chromatin compaction in eukaryotic model organisms through genetic manipulation of H2A should facilitate further experiments to uncover these processes.

## Materials and methods

### Strains and media

The yeast strains used in this study are listed in *Supplementary file 1A*. Yeast cells were grown in YPD at 30°C unless otherwise noted. C-terminal tagging of yeast proteins was performed as described previously (*Longtine et al., 1998*). Mammalian cell lines were maintained at 37°C and 5% $CO_2$ and cultured with 10% fetal bovine serum and DMEM (Life Technologies, Grand Island, NY).

### Histone sequence database construction and analysis

Sequences were initially extracted from the Entrez database using a keyword search for 'histone', and removing non-histone sequences by using keyword searches such as 'histone-like', 'ubiquitin', and 'acetyl', yielding 54,646 results. Blast 2.0 (*Camacho et al., 2009*) was used to align the sequences against the highly conserved histone fold region of the four core histones from *Homo sapiens*. Thresholds for true hits were set at >35% identity match and >90% overlap match with the histone fold globular domain region. All duplicate sequences were removed, and further sequence comparisons were made for histone H3 and H2A sequences to filter variants within them. The canonical sequence data sets comprised 672 sequences for histone H3, 357 sequences for histone H4, 518 sequences for histone H2B, and 435 sequences for histone H2A. To further select one canonical sequence for a species among isotypes and variants when annotation was missing, the sequences were compared to the canonical *H. sapiens* and *S. cerevisiae* sequence, and the sequence with the highest similarity was selected. Using only completely sequenced species, the final histone sequence data set included canonical sequences for 160 species from plants, fungi, protozoa, and animals, with genome sizes ranging from 8 to 5600 Mbp.

Sequences for the four core histones were subsequently split into the N-terminal tail, globular domain, and C-terminal tail (in the case of H2A and H2B) sub-sequences. For discovery of patterns of residue changes according to genome size, each of the sub-sequences was further sub-grouped into small (<100 Mbp), medium (100–1000 Mbp), and large (>1000 Mbp) genome sizes. The frequency of the amino acid residues in each sequence in the sub-groups was determined, and a p-value for the comparison between sub groups was obtained using a Mann–Whitney U Test. Multiple sequence alignment profiles were created using the Muscle sequence comparison tool from Embl-EBI (*Edgar, 2004a, 2004b*). Weblogo3 (*Schneider and Stephens, 1990*; *Crooks et al., 2004*) was used for motif discovery. Heat maps for residue positions were constructed using Cluster 3.0 (*de Hoon et al., 2004*) and Java Treeview (*Saldanha, 2004*).

### Yeast H2A mutagenesis

Site directed mutagenesis was performed using the QuickChange Lightning kit (Agilent Technologies, Santa Clara, CA) on the pFL142 plasmid. *Supplementary file 1B* contains all the plasmids that were used and constructed in this study. The sequences of primers are listed in *Supplementary file 1C*. The correct mutation was verified by sequencing.

### Measurement of yeast nuclear volume

Yeast strains were generated that contained a C-terminally tagged Nup49p-GFP fusion. Cells were grown in a rich medium to 0.6–0.8 × $10^7$ cells/ml, fixed in a growth medium with 4% paraformaldehyde for 15 min at room temperature, washed twice in PBS, and mounted on a poly-L-lysine-coated slide with mounting medium (Vector Laboratories, Burlingame, CA). Z-stacks were obtained as described in the microscopy imaging section, and GFP excitation was achieved at 488 nm. Resulting z-stack images were de-convolved using a constrained iterative algorithm from SlideBook 5.0 software and nuclear volumes were measured by masking the inside of each nucleus, which were delineated by the GFP signal. The resulting mask was used to calculate volumes through the SlideBook software. Statistical analysis was performed using the Student's *t* test.

## Measurement of yeast cellular volume

Yeast strains were grown in rich medium to $0.6–0.8 \times 10^7$ cells/ml, fixed in growth medium with 4% paraformaldehyde for 15 min at room temperature, washed twice in PBS, and stained with a 1:50 dilution of concanavalin A conjugated with tetramethylrhodamine (Life Technologies) for 15 min at room temperature. Cells were washed twice in PBS, once in water, and mounted on a poly-L-lysine-coated slide with mounting medium. Z-stacks were obtained as described in the microscopy imaging section with mRFP excitation. Cell volume was measured by masking the inside of the RFP signal as described in the measurement of yeast nuclear volume.

## FISH probes

For yeast FISH analysis, DNA templates for probes 1, 3, and 4 came from cosmids 71042, 70912, and 70982 (American Type Culture Collection, Manassas, VA) as described elsewhere (*Guacci et al., 1994*). DNA templates for Probe 2 were obtained by PCR amplification of a 10-kb region starting at coordinate 364647 of chromosome 16 using three primer pairs (Probe2_P1, Probe2_P2, Probe2_P3, *Supplementary file 1C*). All DNA templates were digested to smaller fragments using Sau3a (New England BioLabs, Ipswich, MA). Fragments were directly labeled using BioPrime labeling kit (Life Technologies) with either ChromaTide Alexa Fluor 488-5-dUTP or ChromaTide Alexa Fluor 568-5-dUTP (Life Technologies).

For human cell FISH analysis, DNA templates for probes came from BACS RP11-252L24 and RP11-195J4 spaced 0.488 Mb apart on chromosome 1. Each BAC was digested into smaller fragments using Sau3a and fragments were directly labeled using BioPrime labeling kit with either ChromaTide Alexa Fluor 488-5-dUTP or ChromaTide Alexa Fluor 568-5-dUTP, as described above.

## Fluorescent in situ hybridization analysis in yeast

Yeast strains were grown in rich medium to $0.6–0.8 \times 10^7$ cells/ml and fixed in a growth medium with 4% paraformaldehyde for 15 min at room temperature. Cells were then washed twice in the growth medium and re-suspended in 2 ml of EDTA-KOH (0.1 M, pH 8.0) and 10 mM DTT and incubated for 10 min with shaking at 30°C. Cells were spun down and re-suspended in 2 ml of YPD + 1.2 M sorbitol with 50 µg/ml of Zymolyase 100-T (Sunrise Science Products, San Diego, CA) and 400 U/ml of lyticase (Sigma-Aldrich, St. Louis, MO) and incubated at 30°C for 16 min with shaking. Spheroplasts were then washed twice in YPD + 1.2 M sorbitol and transferred to a poly-L-lysine-coated slide. After settling for 5 min, excess liquid was aspirated away and the slides were allowed to air dry for 5 additional min. The slides were washed in methanol for 10 min and then acetone for 30 s before air drying. Cells were then dehydrated in a series of cold ethanol washes (70%, 80%, 90%, 100%, 1 min each) and allowed to air dry. Denaturing solution (70% deionized formamide, 2 × SSC) was added to the slide, and cells were denatured at 75°C for 7–10 min. The slides were immediately put through another cold ethanol dehydration series and allowed to air dry. Hybridization solution (50% deionized formamide, 2 × SSC, 10% dextran sulfate, 100 ng/µl salmon sperm DNA) containing fresh probes was added to the slide, and the probes were hybridized for 40–48 hr at 37°C. The slides were then washed in two 5 min washes in 0.05 × SSC at 48°C and washed twice in BT Buffer (0.15 M NaHCO3 pH 7.5, 0.1% Tween) for 5 min at room temperature. Mounting medium containing DAPI (Vector Laboratories) was added to the slides, and a coverslip was sealed with nail polish.

Inter-probe distances were measured in single projections as described elsewhere (*Bystricky et al., 2004*) by finding the pixel distance between weighted centers of the green signal and red signal and converted to nm by the appropriate factor.

## Microscopy imaging

A 3i Marianas SDC confocal microscope equipped with a Zeiss AxioObserver Z1 with a 100 × /1.45 NA objective and Yokogawa CSU-22 confocal head was used. Images were captured by a Hamamatsu EMCCD C9100-13 camera controlled by Slidebook 5.0/5.5 (Intelligent Imaging Innovations, Denver, CO). DAPI, GFP, mRFP, and Far-red images were acquired by excitation at 360 nm, 488 nm, 561 nm, and 640 nm from a high-speed AOTF laser launch line. A step size of 0.3 (yeast) or 0.5 (human) µm was used for z-stack acquisition.

## Micrococcal nuclease digestion

Micrococcal nuclease (MNase) digestions were performed on exponentially growing yeast cells as described previously, except that the enzyme was obtained from Sigma-Aldrich (Sigma-Aldrich) (*Rando, 2010*).

## RNA expression analysis

RNA was extracted from exponentially growing yeast as described previously (*Schmitt et al., 1990*). PolyA-RNA was prepared, labeled, and hybridized to Affymetrix Gene ChIP Yeast Genome 2.0 array by the UCLA clinical microarray core facility and data normalized according to manufacturer's indications. The data are accessible at Gene Expression Omnibus with accession number GSE50440.

## DNA template and histone preparation for in vitro studies

A plasmid containing 12 tandem 177 bp repeats of the high affinity 601 sequence was obtained from Craig L Peterson's laboratory (*Shogren-Knaak et al., 2006*). DNA arrays were prepared as described previously (*Luger et al., 1999*). After excision with EcoRV, the arrays were gel purified. QuikChange Lightning Site-Directed Mutagenesis (Agilent Technologies) was used to create H2A ΔR11 using primers as listed in *Supplementary file 1C*. Recombinant *X. laevis* histones were expressed in bacteria and purified as described previously (*Luger et al., 1999*). Equimolar amounts of all histones were co-folded to form octamers. Intact octamers were purified from aggregates and free H2A-H2B dimers using Pharmacia Superdex 200 gel filtration column (GE Healthcare Bio-Sciences, Pittsburgh, PA).

## Nucleosome array assembly

Recombinant histone octamers and the 601-177-12 DNA template (*Lowary and Widom, 1998*) were combined in stoichiometric amounts where 1.0 equivalent of histone octamers and 1.0 equivalents of DNA template were mixed in 2.0 M NaCl. Nucleosome arrays were assembled by step-wise salt dialysis in decreasing NaCl concentration: 1.6 M, 1.2 M, 1.0 M, 0.6 M, 0.4 M, 0.1 M, and 0.025 M (in 10 mM Tris pH 8.0, 0.25 mM EDTA), followed by exchanges with 2.5 mM NaCl and 10 mM Tris pH 8.0 without EDTA. Each dialysis step was performed at 4°C for 4 hr to overnight. Partially assembled chromatin was eliminated by precipitation in 4.0 mM $MgCl_2$ (*Dorigo et al., 2003*). The extent of array saturation was assessed by ScaI digestion (200 ng total DNA/chromatin, 3 units ScaI, 50 mM NaCl, 50 mM Tris pH 7.4, 0.5 mM $MgCl_2$), performed for 16 hr at room temperature followed by 1 hr at 37°C, and subsequent analysis using a 5% native polyacrylamide gel (*Luger et al., 1999*).

## Analytical ultracentrifugation

Nucleosome arrays were allowed to equilibrate at room temperature in buffer (2.5 mM NaCl, 10 mM Tris–HCl pH 8.0) containing either 0.1 mM EDTA or 0.6 and 0.8 mM $MgCl_2$. Samples were centrifuged at 20,000 RPM on a Beckman Optima XL-I analytical ultracentrifuge using an An60 Ti rotor after a 1 hr equilibration at 20°C under vacuum. Time-dependent sedimentation was monitored at 260 nm. Boundaries were analyzed by the method of van Holde and Weischet (*Weischet et al., 1978*; *Hansen and Turgeon, 1999*).

## Combined immunofluorescence and fluorescent in situ hybridization in human cells

N-terminally HA-tagged WT H2A of *X. laevis* was cloned by PCR into mammalian expression vector pCMV-HA (Clontech Laboratories, Mountain View, CA) between EcoRI and NotI sites. C-terminally Myc-FLAG-tagged human H2A, in a mammalian expression vector, was obtained from OriGene (RC200688, Origene Technologies, Rockville, MD). Site directed mutagenesis was performed using the QuickChange Lightning kit (Agilent Technologies) on these expression plasmids. Human cells (HEK293, IMR90 and MDA-MB-453) were grown on glass coverslips in 24-well plates in DMEM containing 10% fetal bovine serum and transfected with the indicated H2A expression plasmids using BioT transfection reagent (Bioland Scientific, Paramount, CA) or Lipofectamine LTX with Plus reagent (Life Technologies). Cells were grown for 48 hr post-transfection. For immunofluorescence only, transfected cells were fixed with ice-cold methanol for 15 min at −20°C followed by washing with PBS-T. For combined immunofluorescence and FISH, transfected cells were fixed with 4% paraformaldehyde in PBS for 10 min at room temperature followed by washing with PBS. Cells were then permeabilized in 0.5% Triton X-100 in PBS for 10 min at room temperature followed by washing with PBS. Cells were blocked in 5% BSA and incubated with anti-HA antibody (ab9110; 1:250 dilution, Abcam, Cambridge, MA) or anti-FLAG antibody (F1804; 1:1000 dilution, Sigma-Aldrich). Cells were washed and incubated with secondary antibody (A11008; 1:500 Alexa Fluor 488 goat anti-rabbit, A21245; 1:250 Alexa Fluor 647 goat anti-rabbit, A11001; 1:500 Alexa Fluor 488 goat anti-mouse, or A21235; 1:100 Alexa Fluor 647 goat anti-mouse, Life Technologies). For immunofluorescence, cells were washed and then incubated with Hoechst stain (0.001 mg/ml in PBS). After final washes, cover slips were mounted and imaged. Fluorescence was visualized as above except with the use of 63X magnification.

For FISH, cells were washed, following secondary antibody incubation, in CSK buffer (100 mM NaCl, 300 mM sucrose, 3 mM MgCl$_2$, 10 mM PIPES pH 6.8) and permeabilized in CSKT buffer (CSK+0.5% Triton X-100) before being fixed for 10 min in 4% paraformaldehyde in PBS at room temperature. Cells were immediately put through a cold ethanol dehydration series (5 min each at 85%, 95%, and 100%) and allowed to air dry. Cells were rehydrated in 2 × SSC for 5 min and then RNase-treated for 30 min at 37°C in a humid chamber. Cells were washed with 2 × SSC and denatured at 80°C for 15–20 min with 70% deionized formamide and 2 × SSC. They were immediately cooled with cold 2 × SSC and put through another cold ethanol dehydration series. Probes were added to cells and allowed to hybridize for 48 hr. After hybridization, cells were washed with 50% formamide in 2 × SSC, 2 × SSC, and 1 × SSC containing DAPI. Slides were mounted, imaged, and analyzed as described above. Nuclear staining, in H2A-expressing cells, was used to measure lengths of the long and short orthogonal nuclear axes. Estimated nuclear cross-sectional area was calculated using the following formula: Area = (D$_1$/2) * (D$_2$/2)*π, where D$_1$ and D$_2$ are long and short axis lengths, respectively.

## Competition assays

Two sets of yeast strains were generated in which Pgk1p was C-terminally fused with either GFP or RFP (*Supplementary file 1A*). GFP-labeled WT H2A strains were co-cultured with RFP-labeled mutant H2A strains at a 1:1 ratio and at an optical density of ~0.4. Corresponding co-cultures with switched fluorescent labels were also made. Cultures were incubated at 30°C for 72 hr and were diluted every 6–12 hr to maintain cells in exponential growth phase. Samples were collected every 12 hr for analysis by flow cytometry. Collected cells were fixed in 70% ethanol, washed, and re-suspended in 50 mM sodium citrate, pH 7.0, and mildly sonicated to disrupt aggregates. GFP- and RFP- labeled cells were counted using a Becton Dickinson FACScan cytometer, and the proportion of each in the population was calculated.

## Cell cycle analysis

Cell cycle analysis of exponentially growing cells was performed essentially as described previously (*Zou et al., 1997*), except that cells were stained with 1 μM SYTOX Green (Life Technologies).

## Spot tests

Approximately 1.0 × 10$^7$ exponentially growing yeast cells were collected and re-suspended in 100 μl of H$_2$O and 10-fold serially diluted. Subsequently, 5 μl was spotted on agar plates containing media and drugs as indicated in the figures and incubated at 30°C for 2–6 days.

## Acknowledgements

We acknowledge the support of Martin Phillips and the UCLA-Department of Energy Biochemistry Instrumentation Facility for analytical ultracentrifuge experiments, the Jonsson Comprehensive Cancer Center flow cytometry core facility (supported by grants P30 CA016042 and 5P30 AI028697) and the UCLA Broad Stem Cell Center Sequencing Core. We thank Michael Grunstein and Fred Winston for providing yeast strains TSY107 and FY406, respectively. BM was supported partially by a Howard Hughes Medical Institute Medical Student Fellowship and a Philip Whitcome Pre-doctoral Training Program grant. TS was supported partially by a UCLA Cancer Cell Biology Postdoctoral Training grant. This work was funded by an NIH Director's Innovator Award to SKK.

## Additional information

### Funding

| Funder | Grant reference number | Author |
| --- | --- | --- |
| National Institutes of Health | Director's Innovator Award, 1 DP2 OD006515 | Siavash K Kurdistani |
| Howard Hughes Medical Institute | Medical Student Fellowship | Benjamin R Macadangdang |
| University of California | Philip Whitcome Pre-doctoral Training Program | Benjamin R Macadangdang |

| Funder | Grant reference number | Author |
|---|---|---|
| University of California | UCLA Cancer Cell Biology Postdoctoral Training Program | Tanya Spektor |

The funders had no role in study design, data collection and interpretation, or the decision to submit the work for publication.

### Author contributions
BRM, AO, TS, OAC, MV, Conception and design, Acquisition of data, Analysis and interpretation of data, Drafting or revising the article; FS, Acquisition of data, Analysis and interpretation of data, Drafting or revising the article; MFC, SKK, Conception and design, Analysis and interpretation of data, Drafting or revising the article

# Additional files

### Supplementary files
• Supplementary file 1. Tables of yeast strains, plasmids, and primers.

• Supplementary file 2. Table of yeast FISH results.

• Supplementary file 3. Table of yeast nuclear and cellular volumes.

• Supplementary file 4. Table of human FISH results.

• Supplementary file 5. Table of human nuclear area results.

### Major dataset

The following dataset was generated:

| Author(s) | Year | Dataset title | Dataset ID and/or URL | Database, license, and accessibility information |
|---|---|---|---|---|
| Siavash Kurdistani | 2014 | Expression data from Saccharomyces cerevisiae histone H2A mutants | http://www.ncbi.nlm.nih.gov/geo/query/acc.cgi?acc=GSE50440 | Publicly Available at NCBI Gene Expression Omnibus. |

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
