## [Decision Letter]

Thank you for sending your work entitled “Evolution of Histone 2A for Chromatin Compaction in Eukaryotes” for consideration at *eLife*. Your article has been favorably evaluated by James Manley (Senior editor) and 3 reviewers, one of whom is a member of our Board of Reviewing Editors.

The Reviewing editor and the other reviewers discussed their comments before we reached this decision, and the Reviewing editor has assembled the following comments to help you prepare a revised submission.

This paper explores a novel and interesting hypothesis, which is that the inclusion of extra arginine residues at position 3 and 11 in the N-terminal tail of histone H2A allows a greater degree of compaction of the chromatin. The authors found a strong correlation between the number of arginine residues in the N-terminal domain (NTD) of histone H2A and genome size. This observation led them to postulate that the NTD arginines promote chromatin compaction and a consequent reduction in the nuclear volume. A variety of substitution, insertion, and deletion mutations were constructed in the NTDs of yeast and human H2A. A good correlation between NTD arginines and chromatin compaction was observed. There was not, however, a strong correlation between chromatin compaction and nuclear volume. This study was based on an innovative idea and contains information that will be of interest to chromatin biologists. This work could be appropriate for publication in *eLife* if the following points are suitably addressed.

1) Figure 2 is flawed and should be deleted as a main figure. Figure 2 shows the locations of R3 and R11 on an image of two nucleosomes stacked against each other, to infer that these amino acids are involved in interactions between nucleosomes to promote chromatin compaction. However, the structure used (Davey et al., 2002) was a mononucleosome. This paper itself did not show anything more than a mononucleosome, and any potential nucleosome-nucleosome interactions that have been inferred from the coordinates of the mononucleosome may be crystal-packing artifacts. Furthermore, this modeling should be done using instead Tim Richmond's tetranucleosome structure, albeit that the nucleosome interactions in that structure have not been validated experimentally. As such, Figure 2 does not stand on its own and should be deleted. The models should be remade with two nucleosomes from the tetranucleosome structure and presented as a supplement to Figure 1. If the authors wish to keep the Davey et al. models, the text also needs to acknowledge that the structure that was used to generate the models was a mononucleosome and that the interactions shown have not been validated and may be crystal packing artifacts.

2) The authors should consider and discuss the possibility that the changes in inter-locus distances (i.e., apparent chromatin compaction) may be due to chromatin looping rather than chromatin compaction. This chromatin looping hypothesis could be tested with a 3C-like method, such as Hi-C, but we do not request that such extensive experiments be performed for this paper. This issue will, however, ultimately need to be addressed by the authors.

3) Changes in nuclear volume do not always correlate with chromatin compaction (as summarized in Table 1). Hence, there is not a straightforward relationship between chromatin compaction and nuclear volume. This point needs to be made clearly in the text.

4) There are two issues with Figure 6:

a) The authors reconstituted oligonucleosome arrays with either wild-type H2A or deltaR11. The sedimentation coefficients of the mutant arrays were decreased, suggesting loss of inter-fiber compaction. It should be noted, however that results from the lab of Jeff Hansen (Gordon et al., 2005) showed that reconstituted nucleosome arrays missing only the H2A N-terminal region have a sedimentation profile that is indistinguishable from that of wild-type arrays. Hence, the results from the Hansen lab argue against an essential role of the H2A N-terminal region in chromatin compaction and appear to be inconsistent with the data in Figure 6.

b) The biochemical experiments provide an excellent opportunity to examine the effect of charge on chromatin compaction in the absence of potential histone acetylation. Thus, it could be informative to test the relative effect of R vs. K in these experiments. However, given the ambiguity in the sedimentation data from different labs (as discussed in point 4a, above), it is not certain how useful such experiments would be.

5) Figure 7 shows some mutations of H2A that have been found in cancer. The authors make the argument that the cancer mutations in a single copy of an H2A gene have the potential to affect chromatin compaction, as supported by their overexpression analysis in tissue culture cells. However, in their overexpression study, they show that the mutant histones are expressed to the same level as the endogenous histones. This is the equivalent of a heterozygous mutation in a single copy gene. However, there are 17 canonical H2A genes in humans; hence, the authors are not making a fair argument, as a mutant histone in humans would only be present at 1/34th of the level of the non-mutant histones (assuming equal expression of all histone H2A genes). This fact needs to be made clear in the Discussion.

---

## [Author Response]

*1)*
Figure 2
*is flawed and should be deleted as a main figure.*
Figure 2
*shows the locations of R3 and R11 on an image of two nucleosomes stacked against each other, to infer that these amino acids are involved in interactions between nucleosomes to promote chromatin compaction. However, the structure used (Davey et al., 2002) was a mononucleosome. This paper itself did not show anything more than a mononucleosome, and any potential nucleosome-nucleosome interactions that have been inferred from the coordinates of the mononucleosome may be crystal-packing artifacts. Furthermore, this modeling should be done using instead Tim Richmond's tetranucleosome structure, albeit that the nucleosome interactions in that structure have not been validated experimentally. As such,*
Figure 2
*does not stand on its own and should be deleted. The models should be remade with two nucleosomes from the tetranucleosome structure and presented as a supplement to*
Figure 1*. If the authors wish to keep the Davey et al. models, the text also needs to acknowledge that the structure that was used to generate the models was a mononucleosome and that the interactions shown have not been validated and may be crystal packing artifacts*.

We agree with the comments and have moved Figure 2 to the supplementary material (as a supplement to Figure 1). We have also edited the text to reflect that the internucleosomal contacts are indeed inferred from crystal lattice packing. The reason we performed this analysis in the first place was because the interaction between H4 K16 and the H2A/H2B acidic patch was described initially from crystal lattice packing (Luger et al., 1997).

*2) The authors should consider and discuss the possibility that the changes in inter-locus distances (i.e., apparent chromatin compaction) may be due to chromatin looping rather than chromatin compaction. This chromatin looping hypothesis could be tested with a 3C-like method, such as Hi-C, but we do not request that such extensive experiments be performed for this paper. This issue will, however, ultimately need to be addressed by the authors*.

This is a valid point that we have now explicitly discussed in the paper (see Discussion section). However, we think that this is a less likely possibility because examination of inter-probe distances for four probe-pairs on the same chromosome suggests linear compaction (Figure 2).

*3) Changes in nuclear volume do not always correlate with chromatin compaction (as summarized in Table 1). Hence, there is not a straightforward relationship between chromatin compaction and nuclear volume. This point needs to be made clearly in the text*.

We agree with the reviewers and in fact had attempted to make this point in the discussion of the original submission. We have now further amended our discussion on nuclear volume changes to better emphasize this point.

*4) There are two issues with*
Figure 6*:*

*a) The authors reconstituted oligonucleosome arrays with either wild-type H2A or deltaR11. The sedimentation coefficients of the mutant arrays were decreased, suggesting loss of inter-fiber compaction. It should be noted, however that results from the lab of Jeff Hansen (Gordon et al. 2005) showed that reconstituted nucleosome arrays missing only the H2A N-terminal region have a sedimentation profile that is indistinguishable from that of wild-type arrays. Hence, the results from the Hansen lab argue against an essential role of the H2A N-terminal region in chromatin compaction and appear to be inconsistent with the data in*
Figure 6.

Although we do not have a satisfactory explanation for this discrepancy, we have two counterpoints.

1) We have performed additional in vivo experiments in which we ectopically expressed either WT H2A or an H2A gene missing amino acids 1-12 ( H2A-NTD) in IMR90 cells. Both constructs were FLAG tagged at the C-terminus. As would be predicted from our data, expression of H2A-NTD which is missing R3 and R11 decreased chromatin compaction (increased inter-probe distances) (Figure 4) and also caused an increase in nuclear area (Figure 4). We therefore conclude that deletion of the H2A NTD does indeed affect chromatin compaction in vivo.

2) In Gordon et al., nucleosomal arrays missing the H4 NTD sedimented at a value almost identical to WT octamers: 23.1 S for ΔH4 NTD vs 23.2 S for WT. (Arrays missing H2A sedimented slightly slower than WT arrays with a sedimentation coefficient of 22.7 S). However, other studies have demonstrated that H4 NTD or acetylation of H4K16 do affect chromatin compaction in vitro (Dorigo et al., 2003; Shogren-Knaak et al., 2006).

Some of these discrepancies may be attributable to the nature of assembled chromatin (DNA sequence, method of assembly, etc.) and inherent intricacy of such experiments. But we note that none of the referenced studies have validated their findings *in vivo*. Our *in vitro* data are supported by additional *in vivo* evidence for both ectopic addition of the H2A arginines as well as their removal in yeast and human cell line model systems, respectively.

*b) The biochemical experiments provide an excellent opportunity to examine the effect of charge on chromatin compaction in the absence of potential histone acetylation. Thus, it could be informative to test the relative effect of R vs. K in these experiments. However, given the ambiguity in the sedimentation data from different labs (as discussed in point 4a, above), it is not certain how useful such experiments would be*.

We share the same point of view and in fact did attempt to assemble chromatin in vitro with H2A R11K mutation. But for unknown reasons we were not able to obtain satisfactorily assembled chromatin for sedimentation velocity analysis. However, in addition to the original data on K11 insertion in yeast H2A (Figure 2), we ectopically expressed an H2A gene with R11K mutation in normal human fibroblasts and found no statistically significant effects on chromatin compaction (Figure 4), albeit the average values were slightly larger than WT and trending toward significance (p value=0.02 for FISH distances). Also, R11K did increase the average nuclear area. These data suggest that R11K may have subtle effects on chromatin compaction that would be more apparent if more than one region of the genome was analyzed. Together with the findings from yeast, the data suggest that R11 may be somewhat better than K11 in compacting chromatin and has a greater effect on nuclear area, explaining the evolution of R11 in large genome species.

*5)*
Figure 7
*shows some mutations of H2A that have been found in cancer. The authors make the argument that the cancer mutations in a single copy of an H2A gene have the potential to affect chromatin compaction, as supported by their overexpression analysis in tissue culture cells. However, in their overexpression study, they show that the mutant histones are expressed to the same level as the endogenous histones. This is the equivalent of a heterozygous mutation in a single copy gene. However, there are 17 canonical H2A genes in humans; hence, the authors are not making a fair argument, as a mutant histone in humans would only be present at 1/34th of the level of the non-mutant histones (assuming equal expression of all histone H2A genes). This fact needs to be made clear in the Discussion*.

This is a valid observation and we have now modified the text to better explain this point. We have also performed additional experiments showing that ectopic expression of H2A that bears cancer mutations also affect chromatin compaction (Figure 7).